# In Stent Neo-Atherosclerosis: Pathophysiology, Clinical Implications, Prevention, and Therapeutic Approaches

**DOI:** 10.3390/life12030393

**Published:** 2022-03-08

**Authors:** Annunziata Nusca, Michele Mattia Viscusi, Francesco Piccirillo, Aurelio De Filippis, Antonio Nenna, Cristiano Spadaccio, Francesco Nappi, Camilla Chello, Fabio Mangiacapra, Francesco Grigioni, Massimo Chello, Gian Paolo Ussia

**Affiliations:** 1Cardiology, Università Campus Bio-Medico di Roma, 00128 Rome, Italy; a.nusca@policlinicocampus.it (A.N.); m.viscusi@unicampus.it (M.M.V.); f.piccirillo@unicampus.it (F.P.); a.defilippis@unicampus.it (A.D.F.); f.mangiacapra@unicampus.it (F.M.); f.grigioni@policlinicocampus.it (F.G.); g.ussia@policlinicocampus.it (G.P.U.); 2Cardiac Surgery, Università Campus Bio-Medico di Roma, 00128 Rome, Italy; c.chello@unicampus.it (C.C.); m.chello@policlinicocampus.it (M.C.); 3Cardiac Surgery, Massachusetts General Hospital, Harvard Medical School, Boston, MA 02115, USA; cristianospadaccio@gmail.com; 4Cardiac Surgery, Centre Cardiologique du Nord de Saint-Denis, 93200 Paris, France; francesconappi2@gmail.com

**Keywords:** coronary artery disease, percutaneous coronary intervention, neo-atherosclerosis, in-stent restenosis, optical coherence tomography

## Abstract

Despite the dramatic improvements of revascularization therapies occurring in the past decades, a relevant percentage of patients treated with percutaneous coronary intervention (PCI) still develops stent failure due to neo-atherosclerosis (NA). This histopathological phenomenon following stent implantation represents the substrate for late in-stent restenosis (ISR) and late stent thrombosis (ST), with a significant impact on patient’s long-term clinical outcomes. This appears even more remarkable in the setting of drug-eluting stent implantation, where the substantial delay in vascular healing because of the released anti-proliferative agents might increase the occurrence of this complication. Since the underlying pathophysiological mechanisms of NA diverge from native atherosclerosis and early ISR, intra-coronary imaging techniques are crucial for its early detection, providing a proper in vivo assessment of both neo-intimal plaque composition and peri-strut structures. Furthermore, different strategies for NA prevention and treatment have been proposed, including tailored pharmacological therapies as well as specific invasive tools. Considering the increasing population undergoing PCI with drug-eluting stents (DES), this review aims to provide an updated overview of the most recent evidence regarding NA, discussing pathophysiology, contemporary intravascular imaging techniques, and well-established and experimental invasive and pharmacological treatment strategies.

## 1. Introduction

Coronary artery disease (CAD) remains the leading cause of mortality and morbidity worldwide, albeit with many advances in diagnosis and treatment [1]. Nevertheless, the survival rate of patients with CAD has been continuously improving in the last years due to the development and improvement of revascularization therapies, including percutaneous coronary intervention (PCI) and coronary artery bypass graft (CABG) [2]. However, both treatments could be burdened by the activation of accelerated atherosclerosis, which may occur within months to years and lead to revascularization failure associated with unfavorable long-term follow-up outcomes [3].

Since the introduction of coronary angioplasty about forty years ago, in-stent restenosis (ISR) and de novo neo-atherosclerosis (NA) have been recognized as major causes of long-term PCI failure, together with late stent thrombosis [4]. Pure and early (<1 year from stent implantation) ISR is characterized by the proliferation and migration of vascular smooth muscle cells (VSMCs) leading to the development of significant neointimal hyperplasia [5]. Conversely, the development of a novel fibroatheroma within the stent struts, a process also called neo-atherosclerosis, has raised as a crucial favoring factor to induce long-term coronary complications, including late ISR (>1 year from stent deployment) and very late stent thrombosis [6].

As stent failure persists to be a major concern despite the use of new generation DES, the aim of this review is to provide an update on in-stent neo-atherosclerosis, arguing about pathophysiology and contemporary intravascular imaging techniques, and focusing on potential invasive and pharmacological approaches to prevent and treat this late complication.

## 2. Pathophysiology of Native Atherosclerosis, In-Stent Restenosis and Neo-Atherosclerosis

The underlying pathophysiological mechanisms of ISR and NA are complex and significantly different from those causing native atherosclerosis (Figure 1). Indeed, comprehensive knowledge of all these cellular and molecular pathways are fundamental also to stimulate further research on novel molecular targets to prevent this complication [7].

Coronary native atherosclerosis begins with early adaptive or diffuse intimal hyperplasia in atherosclerosis-prone vessels, usually near branch points [8,9]. These intimal masses, formed by a natural accumulation of smooth muscle cells, in the absence of lipids and macrophage foam cells, develop from birth and are considered a physiological response to blood flow rather than a pathological atherosclerotic process, whereas they could enlarge with advancing age progressing to pathological intimal thickening [8,9]. Afterwards, the invasion of lipid pools and macrophages foam cells determines the formation of early and late fibroatheromas with large necrotic cores [3]. Necrotic cores may develop and expand, sometimes quickly due to intraplaque hemorrhage from leaky vasa vasorum. Furthermore, the fibrous cap above the necrotic core is a critical structural component: active proteases released by infiltrating macrophages could steadily induce a thinning of the fibrous cap [3]. The progressive fibrous cap thinning may result in plaque rupture, the leading cause of acute coronary thrombosis. On the other hand, erosions occur in early fibroatheromas or lesions with pathological intimal thickening, where a luminal surface rich in VSMCs, proteoglycan and hyaluronan matrix induces de-endothelialization and platelet thrombosis [10,11].

In-stent de novo atherosclerosis, or neo-atherosclerosis, is histologically characterized by an accumulation of lipid-laden foamy macrophages with or without a necrotic core and/or calcification within the nascent intima following stent implantation [6]. By the way, there is no communication among neo-atherosclerotic necrotic cores and the underlying native plaque [6]. The most common and earliest lesion of NA is the formation of foamy macrophage clusters, which are usually located in the luminal surface or within the peri-strut area [12]. Subsequently, the accumulation of foamy macrophages within the neointimal layer or on the luminal surface could induce the formation of fibroatheroma [12]. The necrotic core often holds acellular fragments with free cholesterol, with an extracellular matrix (ECM) largely damaged. In addition, extensive hemorrhage with fibrin accumulation could be seen in the necrotic core in NA, originating after fissure or rupture of luminal surface or, alternatively, from leaky vasa vasorum developed in to the adventitial layer near the stent struts [12]. Finally, similarly to native vessel atherosclerosis, foamy macrophages migrated in to the neointimal layer may lead to the formation of fibroatheroma with a thin cap, which could, in turn, result in complications such as in-stent plaque rupture and subsequent thrombosis [3,12]. Moreover, also calcification could be observed within the neointima, particularly involving stents with long-term implant duration. Morphological features of calcification in NA varies widely from microcalcification, due to foamy macrophages or VSMCs apoptosis, to fragmented (>1 mm) or sheet-like calcification (>3 mm) derived from calcification of ECM, collagen and VSMCs [13]. Calcification within the in-stent neointima could occur for both BMS and DES. However, the exclusive feature in DES-related NA is calcification of fibrin, mainly observed in paclitaxel-eluting stents [13]. Conversely, VSMCs proliferation and neointimal hyperplasia without foamy macrophages infiltration are frequently observed after BMS implantation and represent the distinctive feature of BMS-related ISR [14].

A detailed comparison between native atherosclerosis and de novo neo-atherosclerosis is reported in Table 1. As mentioned above, while the atherosclerotic process in native vessels typically materializes through several years, in-stent NA usually occurs in a shorter interval after the PCI and appears to be more common after DES implantation rather than BMS implantation [6]. The processes responsible for in-stent NA are not entirely understood; nevertheless, it could be speculated that stent implantation induces vascular injury with endothelial denudation. The anti-proliferative effects of the eluted drugs, strictly related to DES action and efficacy, induce decreased nitric oxide production and reduced expression of anti-thrombotic molecules, leading to an incomplete and belated regeneration of endothelium [15,16]. Particularly, the physiological barrier function of the endothelium is compromised by altered cell junctions, favoring the migration of lipoproteins within the sub-endothelial area, thus leading to NA development [16]. Of note, since NA after BMS implantation develops earlier than atherosclerosis in native coronary, the stented segment is characterized by an incompetent endothelium anyway, regardless the presence of anti-proliferative mechanisms [17]. 

Furthermore, stent implantation induces local blood flow disorders related to changes in shear stress, leading to the activation of regenerating endothelial cells and the subsequent expression of adhesion molecules (such as ICAM-1, PECAM-1, and VCAM-1) in peri-strut locations [18]. Consequently, monocytes adhere to activated endothelium and migrate into the subendothelial space, converting into macrophage-derived foam cells [19]. DES polymers could also induce chronic inflammation characterized by infiltration of lymphocytes, macrophages, and giant cells, contributing to NA development [20]. In addition, human autopsy analysis showed that restenotic DES reveals higher proteoglycan deposition than restenotic BMS, potentially favoring NA, as proteoglycans mediate lipoprotein retention at the subendothelial space [20]. 

Finally, thrombosis in NA is primarily related to plaque rupture; albeit in-stent erosion is rarely observed in DES and BMS, this complication could occur without the formation of the necrotic or the development of fosamy macrophage clusters, regardless the presence of in-stent restenosis [21,22].

## 3. Clinical Impact of Neo-Atherosclerosis

The impact of both ISR and NA significantly differed among bare-metal stents (BMS), first-generation drug-eluting stents (G1-DES) and second-generation DES (G2-DES). Mainly, ISR occurred early and more frequently after BMS implant due to enhanced neointimal hyperplasia [23], whereas this process is prevented by both G1- and G2-DES because of anti-proliferative in-stent drug release, thus leading to lower rates of ISR [23]. However, this unique advantage in terms of ISR has been partially offset by an increased risk of thrombotic complications, especially with first-generation (G1-DES), due to hypersensitivity reactions to stent polymer and chronic inflammation, inducing malapposition and coronary aneurysms development [24]. On the other hand, the development of new stent platforms with G2-DES (thinner struts, biocompatible durable and biodegradable polymers) has improved clinical outcomes, particularly by decreasing late and very late thrombotic complications [25]. Table 2 reported the incidence of definite very late stent thrombosis according to Academic Research Consortium in BMS, G1-DES and G2-DES [26,27,28,29,30,31,32,33].

Conversely, a higher incidence of de novo neo-atherosclerosis with DES compared with BMS has been observed [23]; of note, this difference did not significantly diverge among first- and second-generation DES [34], negatively affecting the long-term outcome of patients undergoing PCI [35,36,37].

Although the well-established correlation between late ISR and adverse cardiac events, few studies investigated the clinical significance of NA detected by intra-coronary imaging techniques on long-term outcome of patients undergoing PCI. Sumino et al. [38] performed an OCT analysis (median follow up of 4.8 years) on 187 patients undergoing coronary stent implantation, mainly G2-DES. Patients with at least one stent affected by NA reported an increased incidence of MACE (all-cause death, non-fatal MI, non-target vessel revascularization, target vessel revascularization, stent-based adverse events) compared with patients without NA (25% vs. 9%, respectively; *p* = 0.002). Cox-regression analysis showed how NA could independently predict MACE (HR 4.14, 1.58–10.8, *p* = 0.004). In another OCT-guided study, 46 patients with NA were followed using OCT for a median of 46 months [39]. The incidence of MACE (composite of death, myocardial infarction and target lesion revascularization) was significantly higher in the NA group than in the no-NA group (36.9 vs. 9.3 %, *p* value < 0.001). Despite these promising results, a large-scale, randomized study is still lacking in order to elucidate whether early detection of NA could be beneficial to improve long-term outcome after coronary stenting.

## 4. In-Stent Restenosis and Neo-Atherosclerosis: The Pivotal Role of Intra-Coronary Imaging

### 4.1. Intravascular Ultrasound (IVUS)

Although IVUS currently provides proper information on atherosclerotic plaque composition of coronary native vessels, particularly with virtual histology-IVUS (VH-IVUS) [40], several limitations exist to discriminate neo-intimal tissues because of the signal interference from stent metal struts [23]. However, multiple IVUS-guided analyses of neo-intima patterns allowed collecting more accurate data on both natural history and tissue characterization of ISR and NA [41,42]. 

The potential risk of in-stent late lumen loss due to NA was firstly suggested by early angiographic and histopathologic studies, thus without the use of intracoronary imaging techniques. Indeed, BMS-related ISR was firstly considered a benign process caused by a transient chronic inflammation leading to neo-intimal hyperplasia. These studies proposed that ISR, occurring between 6 months and one year after receiving a BMS, was followed by a quiescent period of intimal regression with concomitant luminal enlargement between 2 and 3 years after stent placement [43,44]. However, subsequent studies with longer clinical follow-up elucidated that a late luminal re-narrowing usually occurred beyond 4 years, thus suggesting that ISR may not be as stable as previously thought and that more complex molecular pathways should be underlying this late stent complication [45].

DES substantially reduced ISR burden compared to BMS, whereas the incidence of long-term NA resulted considerably higher in DES (31%) than BMS (16%) [6]. This late catch-up phenomenon has been explained by the initial, transient, and drug-driven suppression of neo-intimal growth, followed by the activation of multiple other mechanisms (incomplete re-endothelization, rapid deposition of lipid-laden macrophages, necrotic core formation) gradually leading to NA-related stent failure [23]. Interestingly, IVUS-guided analyses appear to be consistent with angiographic and histopathological findings confirming that stents developing late ISR (either BMS or DES) have neo-intimal composition, with necrotic core and dense calcium thus suggesting in-stent NA [42]. Additionally, in order to assess potential differences of IVUS-based tissue characterization of restenotic neo-intima between BMS and DES, Ando et al. found that patients receiving DES present a significantly higher percentage of lipids and a significantly smaller percentage of fibrous tissue in restenosis lesions compared with those after BMS [14].

### 4.2. Optical Coherence Tomography (OCT)

In order to overcome the previously mentioned technical limitations of IVUS, optical coherence tomography (OCT) has become in the last decade the method of choice to assess distinct morphological characteristics of restenotic tissue (structure, backscatter, microvessels), as well as peri-strut NA composition (degree of macrophage infiltration and lipid deposition, in-stent calcification, fibrous cap thickness, neointimal rupture), as the consequence of a significantly higher resolution capacity (10–20 µm) compared with IVUS (80–120 µm) [46].

On OCT images, Gonzalo et al. reported in 2009 the first in vivo morphological classification of different ISR patterns. According to this, restenotic tissue structure was defined as either layered, homogeneous or heterogeneous [47]. However, despite the widespread adoption of this classification in daily clinical practice and for research purposes, some ISR patterns are not adequately categorizable using this system [48,49], whereas it does not allow an appropriate classification for NA. For these reasons, Yamamoto et al. suggested modifying the OCT-guided ISR classification as follows: homogeneous high-intensity tissue (type I), heterogeneous tissue with signal attenuation (type II), heterogeneous speckled tissue (type III), mixed tissue containing poorly delineated region with an invisible strut (type IV), mixed tissue containing sharply delineated low-intensity region (type V) and bright protruding tissue with an irregular surface (type VI). Additionally, the potential clinical significance of this classification has been further explored; the authors found that the incidence of stent fracture was significantly higher in both type I and IV, whereas the duration between stent implantation and ISR resulted significantly longer in types IV and VI [50].

Interestingly, significant variability of ISR pattern among different stent types has been described, thus supporting the hypothesis that the underlying neo-atheroma composition is specifically different between BMS, G1-DES and G2-DES. Layered and heterogeneous patterns result prevalent with DES; conversely, lesions were homogeneous after BMS implantation [51]. In addition, a strict association between morphologic OCT characteristics and angiographic progression patterns of ISR has been described after DES implantation. In particular, a heterogeneous tissue morphology was more frequently observed in those patients who developed significant late ISR with poor disease progression in the early phases after DES implantation (the so-called jump-up progression). At the same time, homogenous patterns resulted more prevalent among patients with gradual progression of ISR [52].

Moreover, neo-atherosclerotic plaque tissue varies within the DES population. In this regard, after first-generation DES implantation, lipid length, lipid arc degree and the prevalence of a 360° lipid arc seem to be significantly raised, while the fibrous cap was thinner compared with second-generation DES. Therefore, the higher prevalence of these unstable features in the neo-intimal composition of G1-DES might explain the higher incidence of late stent thrombosis with these devices and the reduced frequency of the same phenomenon with G2-DES [53]. Moreover, although stent malapposition and uncovered struts represent the two traditional major causes of late DES thrombosis, multiple OCT analyses elucidated the crucial pathogenetic role of NA in late stent failure due to thrombus formation. Accordingly, a recent OCT-based study enrolling consecutive patients with late DES thrombosis showed a remarkable prevalence of NA (43.3% of the population), with in-stent plaque rupture being the most dominant mechanism of stent thrombosis [54]. Additionally, among stable patients with progressive ISR, some evidence supporting a NA prevalence of about 30% have been reported at long-term follow-up [55,56]. Of note, despite large clinical trials assessing the effect of neo-intimal classification on long-term clinical outcomes are currently lacking, a recent contribution by Kim et al. suggested that among different patterns of NA, the heterogeneous lesions appear to be linked with poorer MACE [57].

In conclusion, in most PCI settings, these intracoronary techniques, OCT or IVUS, might improve procedural success and prevent late stent failure avoiding suboptimal device deployment. However, current guidelines strongly suggest their use, preferentially OCT, to investigate the mechanisms of stent failure [2,58]; increasing awareness and understanding of NA mechanisms, they might potentially guide appropriate invasive treatments. Table 3 reports the most important studies investigating the usefulness of IVUS and OCT in the NA setting [14,42,47,54,55,56,59,60,61,62,63,64]. Figure 2 shows in-stent NA detected by IVUS (panel A) and OCT (panel B).

## 5. Prevention and Treatment of Neo-Atherosclerosis

### 5.1. Pharmacological Strategies

#### 5.1.1. Lipid-Lowering Therapy

The potential interaction between neo-atherosclerosis and low-density lipoprotein cholesterol (LDL-c) levels remains controversial [65], and the current evidence mainly comes from observational OCT-based studies. Kuroda et al. supported that high LDL-c blood concentration may be independently related to NA progression [39]. Recently, small dense LDL-c and malondialdehyde-modified LDL were found significantly higher in patients with NA compared with those without this late stent complication [66]. Oppositely, a sub-study of the EXAMINATION trial suggested that NA resulted strictly associated with stent length rather than the amount of circulating LDL-c [67]. Finally, several data showing potential benefits of a synergistic statin and eicosapentaenoic acid (EPA) therapy to prevent late stent failure confirm the potential relationship between lipid levels and NA [68]. In this regard, the LINK-IT trial, a prospective, randomized, single-center OCT-based study, showed that intensive combined lipid-lowering therapy (high-dose statins with EPA) suppressed NA progression compared to standard therapy (low-dose statins alone), translating in greater minimum lumen area, reduced in-stent lipid progression with macrophage accumulation, and lower target lesion revascularization [69]. Despite these promising findings, there are no specific therapeutic guidelines recommending lipid-lowering therapy to reduce neo-atherosclerosis burden; furthermore, whether the current proposed LDL-c levels targets for secondary prevention (55 mg/dL or 1.4 mmol/L) might prevent NA and the occurrence of stent failure is uncertain. Along these lines, a recent small OCT study reported a lower incidence of NA in patients with LDL-c < 70 mg/dL [70].

#### 5.1.2. Antiplatelet Therapies

The inhibition of platelet activation is a cardinal point of secondary prevention after DES implantation [71]. The European Society of Cardiology (ESC) guidelines emphasize the importance of this strategy and suggest a personalized duration of dual antiplatelet/anticoagulant therapy based on comorbidities and ischemic/bleeding risk [2,72]. The DAPT trial showed that long-term treatment with dual antiplatelet therapy (DAPT) reduced the risk of myocardial infarction and stent thrombosis in patients receiving DES at the cost of a raised risk of moderate bleeding [73]. Interestingly, in this study, patients receiving G1-DES reported the greater reduction in ischemic events following longer DAPT duration [74]. Conversely, in patients receiving G2-DES, a prolonged DAPT resulted in a poorer benefit [74]. Otherwise, the thrombotic risk after PCI is influenced by several clinical (diabetes, acute coronary syndromes, chronic renal failure) and procedural features (multiple stents, stent length, small vessels, malapposition). Thus, the type and duration of antiplatelet strategy should be tailored to the patient’s and procedural characteristics other than the stent type in order to prevent stent thrombosis and avoid excessive bleeding risk. Neo-atherosclerosis is a crucial substrate of late stent thrombosis, which is observed significantly earlier after DES (mean ~420 days) than after BMS (mean ~2160 days), whereas no significant difference between G1 and G2-DES has been noticed since both employ mTOR inhibitors [75]. Nevertheless, no specific recommendation regarding the type and duration of antiplatelet therapy exists for patients with evidence of NA to prevent stent thrombosis. Furthermore, few studies examined the long-term use of P2Y12 inhibitors after coronary stenting. Xie et al., retrospectively enrolling 57,900 patients undergoing PCI, found that among those who continued P2Y12 inhibitors intake, a significant reduction in major cardiovascular events was observed at 5 years follow-up [76]. Similarly, in the THEMIS Trial, long-term therapy with ticagrelor in addition to aspirin reduced cardiovascular death, myocardial infarction, and stroke, although with an increased risk of major bleedings at a median follow-up of almost 40 months [77,78].

Finally, regarding the potential role of novel anticoagulants, the combination of rivaroxaban and aspirin in comparison with aspirin alone did not impact the rate of definite stent thrombosis in patients chronic coronary syndromes with a previous PCI in the COMPASS-PCI trial [79].

#### 5.1.3. Targeting Inflammation and Oxidative Pathways

An activated inflammation is at the base of NA development and progression; thus, patients with documented neointimal hyperplasia at quantitative OCT revealed higher levels of high sensitivity C-reactive protein (hs-CRP) and eosinophil cationic protein levels compared with those without in-stent complications, with a proportional relationship between neointimal burden and systemic levels of inflammatory markers [80]. According to these findings, inflammatory pathways may be potentially targeted, providing new opportunities for IRS and NA prevention and treatment. 

In this regard, the results of the pivotal CANTOS trial demonstrated for the first time that pharmacological strategies explicitly targeting the interleukin-1β (IL-1β) inflammatory pathway may improve cardiovascular outcomes in patients with a history of myocardial infarction and enhanced inflammatory response (hs-CRP > 2 mg/dL) [81]. Of note, patients with the highest clinical benefit were more frequently treated with PCI before randomization, suggesting these anti-inflammatory drugs might have a beneficial effect in PCI setting, attenuating atherosclerosis progression and stent failure [81]. Methotrexate has also been investigated as a preventive strategy for NA development due to its properties to block IL-1 receptors and inhibit the release of other inflammatory cytokines [82]. In a pre-clinical OCT-based study analyzing rabbits treated with DES, the administration of methotrexate led to thinner and smaller lipid-rich neointimal areas [83]. Furthermore, colchicine is another well-known drug with anti-inflammatory and anti-proliferative effects [84]. Many studies investigated its role in the prevention ISR, with contrasting results. In an animal placebo-controlled randomized trial, colchicine significantly reduced late lumen loss compared with placebo [85]. Additionally, in another double-blinded randomized clinical trial, 196 patients undergoing PCI with BMS implantation reported a lower incidence of angiographic and IVUS-defined ISR after treatment with colchicine 0.5 mg twice daily than controls [86].

Finally, recent studies proposed a crucial vaso-protective role of the glucagon-like peptide 1 (GLP-1). The activation of the GLP-1 receptor has been demonstrated to induce the proliferation of endothelial cells, stimulate NO production, and inhibit the migration of monocytes and VSMCs proliferation [87,88,89]. These findings, in turn, suggest that both glucagon-like peptide-1 receptor agonists and dipeptidyl peptidase-4 (DPP-4) inhibitors might potentially modulate ISR and NA development. Indeed, the GLP-1 analogue liraglutide showed lower intimal hyperplasia in a pig model after coronary stent implantation [90]. Accordingly, DPP-4 inhibitors eluting stents improved neointima regeneration in vitro and in a diabetic rabbit model [91]. These agents might also reduce PCI-related late complications through well-known anti-platelet properties [92]. Nevertheless, despite these favorable findings, clinical data available so far are conflicting. A recent observational study investigating the effect of incretins therapy on the risk of ISR/ST in a large cohort of diabetic patients receiving DES did not find any significant benefit of this class of agents on stent failure, albeit a significant reduction in cardiovascular mortality [93].

### 5.2. Device-Based Strategies

#### 5.2.1. Drug-Eluting Balloons and Repeat Drug-Eluting Stent Implantation

The RIBS IV (Restenosis Intra-Stent of Drug-Eluting Stents: Drug-Eluting Balloons vs. Everolimus-Eluting Stents) and RIBS V (Restenosis Intra-Stent of Bare Metal Stents: Paclitaxel-Eluting Balloon vs. Everolimus-Eluting Stent) trials showed that NA appears to be the key pathological substrate of a substantial number of ISR lesions undergoing repeat revascularization [94,95]. If the latter demonstrated the safety and effectiveness of both DEB and DES in treating patients with BMS-ISR, the RIBS IV trial showed a reduced need for repeat interventions with everolimus-eluting stent than DEB in the setting of DES-ISR [94,95]. Furthermore, a pooled analysis of these trials studied the potential prognostic impact of OCT-derived NA on the long-term outcomes of patients with ISR requiring repeated PCI [96]. At 3-years follow-up, the rate of MACEs was similar among patients with and without NA (15% vs. 12%, respectively, *p* = 0.93). These results appear to be consistent with those from an observational study by Tada et al. enrolling consecutive patients undergoing PCI for ISR [97]. According to OCT images, among patients with a heterogeneous pattern of neo-intima suggesting an underlying NA, no increase in re-ISR or repeat target lesion revascularization (TLR) rates was reported at 6–8 months of follow-up, dispelling the concept of a negative predictive role of NA in the setting of repeat percutaneous revascularization. However, these limited data come from small studies with a relatively short angiographic follow-up. Further studies are needed to investigate the real predictive role of NA on hard clinical endpoints in the long-term follow-up of patients receiving coronary stenting and to evaluate the best therapeutic strategy to adopt in this setting.

#### 5.2.2. Debulking Strategies

Not infrequently, NA requires debulking strategies due to the high degree of calcium involving neo-intima [98]. Several small studies using rotational atherectomy, including one randomized trial, suggested that debulking techniques were associated with a lesser amount of residual tissue and a lower rate of target vessel revascularization than plain old balloon angioplasty (POBA) [99]. However, successive trials comparing POBA to rotational atherectomy showed contrasting results with better clinical outcomes in patients undergoing POBA [100]. 

The excimer laser coronary atherectomy (ELCA) has been evaluated in small, randomized studies enrolling patients with ISR [101,102]. Albeit angiography showed a higher acute luminal gain among patients undergoing ELCA, no significant difference in clinical outcomes and target vessel revascularization has been observed in patients undergoing combined treatment (ELCA and POBA or DEB) than those treated with balloon dilation alone [101,102]. 

Notably, suboptimal stent strut expansion due to severe calcification in the intimal and medial layers of the coronary wall represents a major determinant of ISR [103]. In the last years, a novel technique combining a balloon angioplasty catheter with the use of sound waves, the Shockwave intravascular lithotripsy, has emerged as a safe and effective tool for the treatment of severe calcified de novo coronary lesions [104,105]. Its use at the site of coronary narrowing before stent deployment creates microfractures into the calcified tissue without affecting soft tissues, allowing complete stent expansion and apposition. More recently, small series and several case reports employed intra-coronary lithotripsy for the treatment of ISR, in those cases where a severe calcified neointimal tissue was detected by intracoronary imaging or a severe calcification of the coronary wall caused an incomplete stent expansion at the time of the first PCI [106]. Promising short- and mid-term angiographic data were reported in most cases; however, further studies with longer follow-up are needed to confirm the effectiveness of this technique in this complex setting.

#### 5.2.3. Bioresorbable Vascular Scaffolds

In the context of patients suffering multiple ISR requiring further stent implantation, potentially increasing the number of metal layers within the coronary arteries, or those in which DEB failed to guarantee a successful PCI on long-term follow-up, bioresorbable vascular scaffolds (BVS) appear an intriguing alternative treatment option. In this regard, despite being currently an off-label strategy evaluated in selected small studies, BVS showed encouraging results regardless of ISR pattern or stent type [107], whereas long-term clinical outcomes are still a matter of debate. A multicenter Italian registry suggested acceptable rates of cardiovascular events on long-term follow-up; at 15 months of follow-up, the composite endpoint including cardiac death, target vessel myocardial infarction and ischemia-driven target lesion revascularization occurred in 9.1% of patients [108]. Conversely, a recent comparison of BVS, DEB (paclitaxel-eluting balloon), and DES re-implantation (everolimus-eluting stent) for ISR treatment showed that BVS have a similar performance compared with DEB but is inferior to DES [109]. Therefore, data supporting the routine adoption of BVS as an alternative tool for ISR treatment are currently lacking.

Figure 3 summarizes potential pharmacological and device-based strategies for NA prevention and treatment. Of note, despite reasonable pathophysiological bases, most of these approaches are in an experimental phase and need to be proved in further studies. 

## 6. Conclusions

Despite improved technologies and materials, PCI is still burdened by a remarkable risk of stent failure mainly driven by de novo neo-atherosclerosis. Whether early ISR results from neointimal hyperplasia, NA is a more complex phenomenon represented by a novel fibro-atheroma developing within the stent struts but separated from the native plaque. The gradual advancements of DES enabled to dramatically mitigate the impact of ISR; however, the incidence of NA did not differ significantly between DES generations. Meanwhile, the incremental adoption of intra-coronary imaging techniques, mainly OCT, contributed to a more extensive understanding of both morphological characteristics of restenotic tissue and peri-strut composition, thus improving procedural success and providing early detection of stent failure, as suggested by current guidelines. Although none of the available pharmacological therapies clearly demonstrated a net beneficial effect in reducing NA burden, during the last decades, several invasive tools (particularly DEB, debulking techniques and, more recently, BVS) allowed to significantly improve mid-term angiographic results. 

Neo-atherosclerosis represents a challenge for interventional cardiologists. Since the increasing use of percutaneous techniques for coronary atherosclerosis treatment in the last years, further studies aiming to identify safer and more effective preventive and therapeutic strategies are needed. However, we have to take in mind that a successful PCI calls for appropriate intra-coronary imaging techniques in the periprocedural phase and a concomitant comprehensive pharmacological strategy in the acute and long-term follow-up in order to prevent stent failure and its deleterious clinical consequences.

## Figures and Tables

**Figure 1 life-12-00393-f001:**
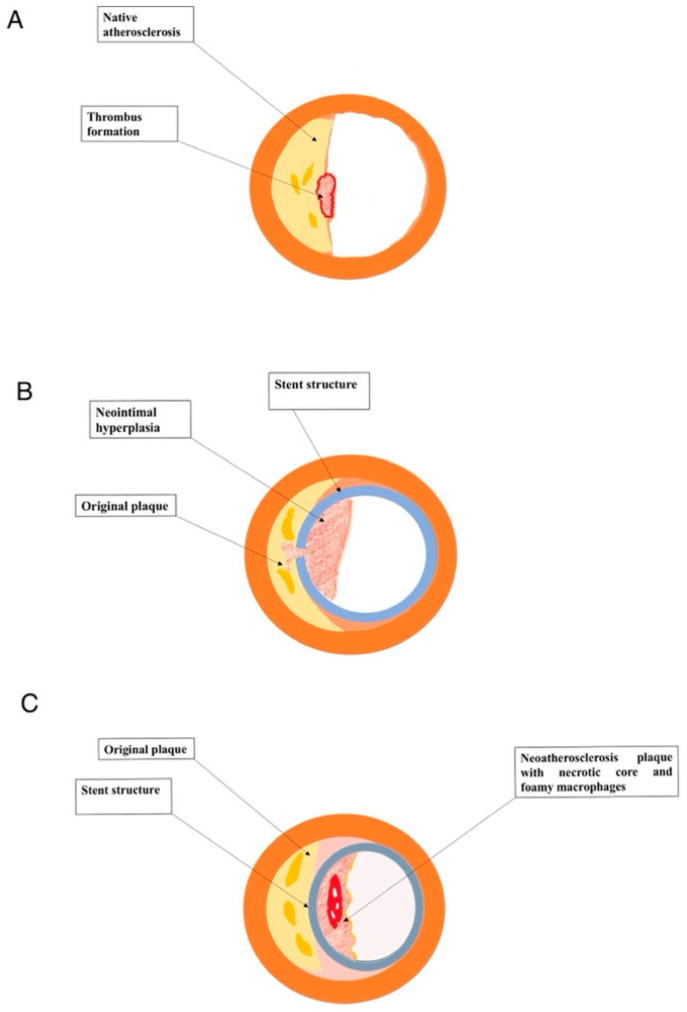
(**A**) Native atherosclerotic plaque. (**B**) In-stent restenosis. (**C**) Intra-stent neo-atherosclerosis.

**Figure 2 life-12-00393-f002:**
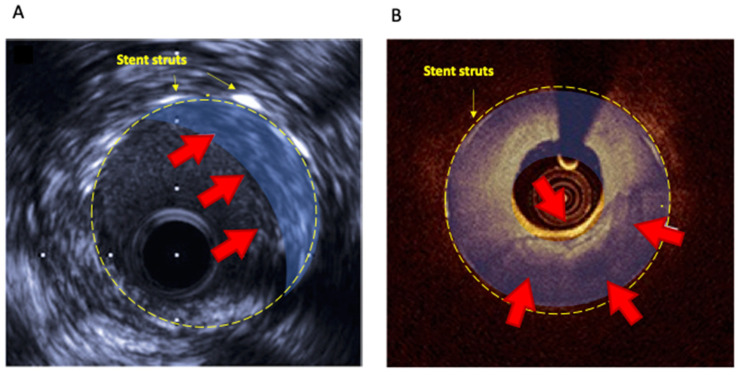
In-stent neoatherosclerosis detected by IVUS (**A**) and OCT (**B**).

**Figure 3 life-12-00393-f003:**
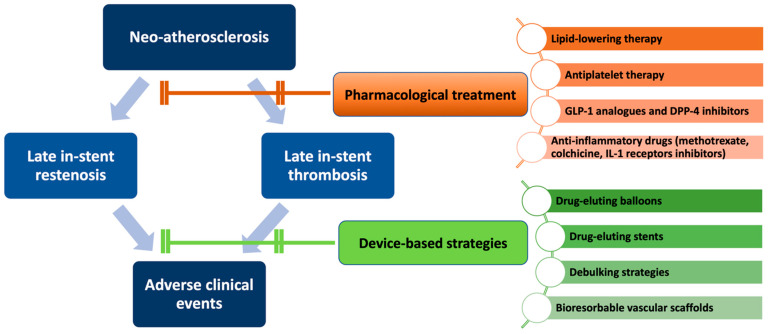
Overview of potential pharmacological and invasive strategies for neo-atherosclerosis prevention and treatment.

**Table 1 life-12-00393-t001:** Profile comparison of native atherosclerosis and neo-atherosclerosis.

	Native Atherosclerosis	Neo-Atherosclerosis
Earliest lesion	Intimal thickening, which could regress	Intimal thickening with VSMCs proliferation
Intimal xanthoma (“fatty streaks”)	Individual foam cells interspersed throughout the intima	Foam cells clusters in surface or in peristrut regions
Pathological intimal thickening	Present	Absent. Rarely present in BMS
Necrotic cores	Deep	Superficial
Intraplaque hemorrhage	Arising from the lumen and/or leaky neoangiogenic vessels	Arising from the lumen and/or leaky neoangiogenic vessels
Plaque erosion	Occasional	Rare
Calcification	Microcalcification, calcified sheets, or calcified fragments	Microcalcification, calcified sheets, or calcified fragments. Calcified fibrin in DES
Eruptive calcified nodules	Rare	Absent
Fibrocalcific plaque	Very common	Common, especially in DES
Thrombosis	60–70% due to plaque rupture; less frequently by plaque erosion. Rarely eruptive calcified nodules.	Primarily due to plaque rupture. In-stent erosion is a rare event
Chronic total occlusion	Very common	Organized thrombus; not always derived from plaque rupture or restenosis

BMS: bare metal stent; DES: drug eluting stent; VSMC: vascular smooth muscle cell.

**Table 2 life-12-00393-t002:** Incidence of definite very late stent thrombosis in BMS, G1-DES and G2-DES.

Study	BMS	G1-DES	G2-DES
Tada et al. [27]	1.5%	2.2%	1.0%
EXAMINATION Trial [28]	2.1%	-	0.8%
TYPHOON Trial [29]	4.0%	3.6%	-
SESAMI Trial [30]	1.3%	1.9%	-
COMFORTABLE-AMI Trial [31]	2.2%	-	3.9%
RACES-MI Trial [32]	-	1.2%	0%
Raber et al. [33]	-	1.6% (PES) 1.0% (SES)	0.3%

BMS: bare metal stent; G1-DES: first-generation drug-eluting stents; G2-DES: second-generation drug-eluting stents; PES: paclitaxel eluting stent; SES: sirolimus eluting stent.

**Table 3 life-12-00393-t003:** Main findings from observational studies on intra-vascular imaging techniques assessing neoatherosclerosis.

Study (Year)	Type of Study	Methodology	Main Findings
Kang et al. (2010) [42]	Observational	VH-IVUS-guided tissue characterization of 117 restenotic lesions after BMS and DES implantation	BMS- and DES-treated lesions develop in-stent necrotic core and dense calcium, suggesting the development of in-stent neoatherosclerosis
Ando et al. (2013) [14]	Observational	IB-IVUS-guided tissue characterization of 54 restenotic lesions after BMS and SES implantation	The neo-intimal tissue after SES implantation had a significantly larger percentage of lipid tissue and a significantly smaller percentage of fibrous tissue compared with that after BMS
Yoshizane et al. (2019) [61]	Observational	IB-IVUS-guided tissue characterization of 125 restenotic lesions after BMS and DES implantation	On long-term FUP, a significant difference was observed in the change of TD of the BMS group (low value in the early period with later increase), whereas TD of the DES group tended to be high from the early period
Gonzalo et al. (2009) [47]	Observational	Quantitative and qualitative OCT-guided restenotic tissue structure characterization of 25 lesions	Layered, homogeneous and heterogeneous pattern of restenotic tissue have been identified.
Habara et al. (2011) [62]	Observational	Quantitative and qualitative OCT-guided restenotic tissue structure characterization of 43 patients with very-late ISR compared with 39 patients with early ISR	Heterogeneous pattern of restenotic tissue was significantly higher in patients with very-late ISR compared to patients with early ISR
Kang et al. (2011) [55]	Observational	OCT-guided analysis of 50 DES in-stent restenosis lesions	OCT findings of NA was frequently identified especially in patients with late ISR, including TFCA-containing neointima, intimal rupture and thrombi.
Kim et al. (2012) [56]	Observational	OCT-guided evaluation of serial changes in stent strut coverage and neointima characteristics of 76 DES-treated lesions at 9-months and 2-years FUP	Neointimal coverage improved during FUP without significant changes in the incidence of malapposed struts and intracoronary thrombus; NA including transformation to lipid-laden neointima increases during extended FUP
Yonetsu et al. (2012) [63]	Observational	Determining the predictors of NA using OCT-analysis of 179 stent-treated lesions	Stent type (DES), stent age (>48 months), age (>65 years), current smoking, chronic kidney disease and ACEi/ARBs use are significant predictors of NA.
Nakamura et al. (2016) [64]	Observational	OCT-guided evaluation of failure mechanisms and NA patterns in 61 patients with very-late DES or BMS stent thrombosis	Uncovered struts, malapposed struts and stent underexpansion were more frequently observed in DES; NA, lipid neo-intima, TCFA neo-intima were more frequently observed and had a more diffuse pattern of distribution in BMS.
Joner et al. (2018) [54]	Observational	OCT-guided assessment of NA in 134 patients with VLST	NA was frequently observed in VLST (43.3%); in-stent plaque rupture resulted the dominant mechanism causing VLST.
Nakamura et al. (2019) [59]	Observational	OCT-guided analysis of NA patterns among 98 patients with ISR and previously BMS/DES treated lesions	NA with ISR was more frequent with DES than BMS and its pattern exhibited a more focal and thicker fibrous cap as compared with BMS.
Yamamoto et al. (2020) [60]	Observational	OCT-guided evaluation of 133 lesions with ISR after DES implantation	Neo-intimal tissue was classified in six different patterns: homogeneous high-intensity tissue (type I), heterogeneous tissue with signal attenuation (type II), speckled heterogeneous tissue (type III), heterogeneous tissue containing poorly delineated region with invisible strut (type IV), heterogeneous tissue containing sharply delineated low-intensity region (type V), or bright protruding tissue with an irregular surface (type VI)

VH-IVUS: virtual histology–intravascular ultrasound; BMS: bare metal stent; DES: drug eluting stent; IB-IVUS: integrated backscatter intravascular ultrasound; SES: sirolimus eluting stent; FUP: follow-up; TD: tissue signal distribution; OCT: optical coherence tomography; ISR: in-stent restenosis; NA: neo-atherosclerosis; TFCA: thin fibrous cap atheroma; ACE-I/ARBs: angiotensin-converting enzyme inhibitors/angiotensin II receptor blockade; VLST: very late stent thrombosis.

## Data Availability

Not applicable.

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
