# Peer review of "In Stent Neo-Atherosclerosis: Pathophysiology, Clinical Implications, Prevention, and Therapeutic Approaches"

_life, 2022, doi:10.3390/life12030393_

Round 1

Reviewer 1 Report

Rows 42-44:

Pure and early (<1 year 42 from stent implantation) ISR is characterized by the migration of significant neointimal 43 hyperplasia caused by vascular smooth muscle cells (VSMCs) proliferation.

Unclear, please rephrase.

Figure 1.

Please denote a stent in figure B.

Table 1.

I think that the timeline of the native atherosclerotic process and neo-atherosclerosis (NA) under stents are not such different as the authors stated, because the data regarding NA are biased by the pathology findings from the people who died, many of them because of the stent failure, and the majority of patients with stents have not such rapid changes. Additionally, patients with stents have already a long history of atherosclerotic development and this is the continuous process on the sick, damaged, atherosclerotic plaque covered with stents. Native atherosclerotic plaques also have periods of accelerated inner development, and in my opinion, the concept of the different timelines between native and NA are at least relative.

Lines: 121-124:

Furthermore, stent implantation induces local blood flow disorders related to changes in shear stress, leading to the activation of regenerating endothelial cells and the subsequent expression of adhesion molecules (such as ICAM-1, PECAM-1, and VCAM-1) in peri-strut locations.

Needs reference.

For this kind of review, it will be necessary to obtain some pathohistological images. If authors have not such an investigation they should ask somebody who has.

Part 3. 135-165 rows.

Since the late and very late stent thrombosis are mainly the consequences of NA and most important clinical adverse event of this process, I suggest to the authors provide some table with the data comparing late and very late stent thrombosis between bare-metal stents (BMS), the first generation of drug-eluting stents (DES) and the second generation of DES. 

Part 4.1. Intravascular Ultrasound (IVUS) from rows 168-198.

Again, in a review article like this in the high-ranking journal, the authors should provide the pictures of NA made by IVUS with descriptions and definitions of NA changes. If authors haven’t their own pictures they should look for permission to use somebody else pictures. But if authors have neither images of their own it is doubtful if they are competent to write this kind of review article.

Part 4.2. Optical Coherence Tomography (OCT) rows 199-250

The same, very important comment, the pictures are missing. They are very important in this kind of review.

General comment about intravascular imaging parts

Provide a table considering the most important IVUS and OCT investigations about NA.

5.1.1. Lipid-lowering therapy rows 252-270

We don’t know whether guidelines recommendation for high-risk cardiovascular events for LDL level (LDL<1,4 mmol/L) can influence very late stent failure. And that is the basic question regarding the lipid-lowering therapy for the NA prevention and late and very late stent failure.

5.1.2. Antiplatelet therapy

The type of stent used for PCI is only one of the factors which can influence the rate of very late stent failure. The complexity of PCI and the importance of stents' position are very important. Now I’m prone to favorite long term dual platelet or use of stronger P2Y12 blockers alone for complex PCI with clinical features of higher risk (diabetes, poor control of LDL, higher Lp(a), smokers, etc), with lower risk for major bleeding (patients younger than 75, and without a history of gastrointestinal or other diseases which could provoke bleeding).

Is there any MS about the long-term use of P2Y12 blockers (longer than 3 years) in patients with stents?

We are witnesses of stent thrombosis after 10 and more years now. The stents are locus resistensiae minoris for life.

5.1.3. Targeting inflammation and oxidative pathways, rows 290-358

This is an un-appropriately long part with full of experimental and small clinical findings regarding prevention of in/stent restenosis, and not NA or very late thrombosis which is the main consequence of NA and the main topic of this review.  

5.2.1. Drug-eluting balloons and repeat drug-eluting stent implantation rows 360-379

The problems for all these trials are a relatively short follow-up, soft endpoints as outcomes, and the relatively small number of patients and we must be very cautious with the interpretation of clinical values of these studies. This critical approach to the results is lacking in the text.

5.2.2. Debulking strategies

As the authors stated, debulking techniques are not widely available and in spite of initially larger angiographic gain this is not reflecting the clinical benefit and the place for this technique is far from clinical significance. Too much space for this.

5.2.3. Bioresorbable vascular scaffolds

BVS should be used only as a part of clinical studies in patients with in-stent restenosis, and we must wait for the next generation of BVS to go ahead.

Figure 2. Almost all these treatment options are lacking evidence-based data for use in this setting and some of them are totally in the experimental phase, it should be stated.

General comment: do we have some investigation with long-term follow-up of stented patients and the incidence of NA in consecutive patients?

In my opinion, this review article is lacking more pictures, tables, and information. Some parts are overwritten and some of them need more data.

Conclusion

The article needs major revision to be considered for publication in life.  

Author Response

We appreciate the positive feedback from the reviewer. We thank him/her for all helpful comments that we believe to have fully addressed in this revised version of the manuscript.

  1. Rows 42-44: Pure and early (<1 year 42 from stent implantation) ISR is characterized by the migration of significant neointimal 43 hyperplasia caused by vascular smooth muscle cells (VSMCs) proliferation. Unclear, please rephrase.

According to the Reviewer’s comment, we rephrased the sentence as reported in the text (page 1-2, lines 42-45): “Pure and early (<1 year from stent implantation) ISR is characterized by the proliferation and migration of vascular smooth muscle cells (VSMCs) leading to the development of significant neointimal hyperplasia.” We also added a specific reference (reference 5 in the revised manuscript, R. Komatsu, M. Ueda, T. Naruko, A. Kojima, A.E. Becker. Neointimal tissue response at sites of coronary stenting in humans: macroscopic, histological, and immunohistochemical analyses, Circulation, 98 (1998), pp. 224-233).

  1. Figure 1. Please denote a stent in figure B.

We denoted the stent in Figure 1B as suggested by the Reviewer. Thank you.

  1. Table 1. I think that the timeline of the native atherosclerotic process and neo-atherosclerosis (NA) under stents are not such different as the authors stated, because the data regarding NA are biased by the pathology findings from the people who died, many of them because of the stent failure, and the majority of patients with stents have not such rapid changes. Additionally, patients with stents have already a long history of atherosclerotic development and this is the continuous process on the sick, damaged, atherosclerotic plaque covered with stents. Native atherosclerotic plaques also have periods of accelerated inner development, and in my opinion, the concept of the different timelines between native and NA are at least relative.

We thank the Reviewer for his/her comment. We agree with him/her regarding the difficulty in standardizing the specific timeline of the native atherosclerotic process and neo-atherosclerosis. However, it is generally accepted, according to several expert pathologists, that native atherosclerosis develops over decades since it progresses from lipid accumulation and intimal thickening (Ross, R. The pathogenesis of atherosclerosis: a perspective for the 1990s. Nature 362, 801–809 (1993); Libby, P. Inflammation in atherosclerosis. Nature 420, 868–874 (2002); Hansson, G. K., Libby, P., Schonbeck, U. & Yan, Z. Q. Innate and adaptive immunity in the pathogenesis of atherosclerosis. Circ. Res. 91, 281–291 (2002); Ikari, Y., McManus, B. M., Kenyon, J. & Schwartz, S. M. Neonatal intima formation in the human coronary artery. Arterioscler. Thromb. Vasc. Biol. 19, 2036–2040 (1999); Yahagi K, Kolodgie FD, Otsuka F, Finn AV, Davis HR, Joner M, Virmani R. Pathophysiology of native coronary, vein graft, and in-stent atherosclerosis. Nat Rev Cardiol. 2016 Feb;13(2):79-98). Indeed, different evolutions of this phenomenon have been recognized: a stable fibrous cap and calcification are generally associated with a long-term clinical course; otherwise, the rapid development of a necrotic core and erosion/rupture of the fibrotic cup might be responsible for the occurrence of acute thrombotic events. We also agree with the Reviewer that most data regarding temporal patterns of neo-atherosclerosis come from autopsy studies of explanted stents. According to these studies, the development of NA occurs within one year after DES and later after BMS (Yahagi K, Kolodgie FD, Otsuka F, Finn AV, Davis HR, Joner M, Virmani R. Pathophysiology of native coronary, vein graft, and in-stent atherosclerosis. Nat Rev Cardiol. 2016 Feb;13(2):79-98; Nakazawa, G. et al. The pathology of neo-atherosclerosis in human coronary implants bare-metal and drug-eluting stents. J. Am. Coll. Cardiol. 57, 1314–1322 (2011); Otsuka, F. et al. Pathology of second-generation everolimus-eluting stents versus first-generation sirolimus- and paclitaxel-eluting stents in humans. Circulation 129, 211–223 (2014)). The early detection compared with native atherosclerosis might be due according to the specific morphological diversity and the presence of a specific trigger such as stent implantation and consequent vascular injury. Based on the concerns regarding the temporal pattern and following the Reviewer’s comment, we remove from Table 1 the line “timeline” avoiding reporting incomplete concepts.

  1. Lines: 121-124: Furthermore, stent implantation induces local blood flow disorders related to changes in shear stress, leading to the activation of regenerating endothelial cells and the subsequent expression of adhesion molecules (such as ICAM-1, PECAM-1, and VCAM-1) in peri-strut locations. Needs reference.

We add a specific reference as suggested by the reviewer (now reference 18 in the revised version of the manuscript): “Chaabane C, Otsuka F, Virmani R, Bochaton-Piallat ML. Biological responses in stented arteries. Cardiovasc Res. 2013;99(2):353-63”.

  1. For this kind of review, it will be necessary to obtain some pathohistological images. If authors have not such an investigation they should ask somebody who has.

We agree with the Reviewer that histological images could significantly improve the comprehension of some pathophysiological concepts in the manuscript. Unfortunately, we do not have histological images since none of us is a pathologist. For this reason, we added schematic representations of native atherosclerosis, ISR and NA as reported in Figure 1. We decided to proceed in this way since the histological description of NA is not the main purpose of our review, and because previous colleagues, mostly pathologists, have already discussed this topic in previous excellent reviews (Eur Heart J. 2015 Aug 21;36(32):2147-59. doi: 10.1093/eurheartj/ehv205; Arterioscler Thromb Vasc Biol. 2015 Oct;35(10):e43-9. doi: 10.1161/ATVBAHA.115.306251; Nat Rev Cardiol. 2016 Feb;13(2):79-98. doi: 10.1038/nrcardio.2015.164; Am J Cardiol. 2018 Jun 1;121(11):1413-1435. doi: 10.1016/j.amjcard.2018.02.017). We hope that the Reviewer could understand our position.

  1. Part 3. 135-165 rows. Since the late and very late stent thrombosis are mainly the consequences of NA and most important clinical adverse event of this process, I suggest to the authors provide some table with the data comparing late and very late stent thrombosis between bare-metal stents (BMS), the first generation of drug-eluting stents (DES) and the second generation of DES.

Thanks for the comment. As suggested by the Reviewer, we added a new table with studies comparing definite very late stent thrombosis between BMS, first-generation DES and second-generation DES (now Table 2 in the revised version of the manuscript). We also included cited studies in the references list (references 26-33 in the new version of the review).

  1. Part 4.1. Intravascular Ultrasound (IVUS) from rows 168-198. Again, in a review article like this in the high-ranking journal, the authors should provide the pictures of NA made by IVUS with descriptions and definitions of NA changes. If authors haven’t their own pictures they should look for permission to use somebody else pictures. But if authors have neither images of their own it is doubtful if they are competent to write this kind of review article.

We agree with the Reviewer regarding the undoubtedly usefulness of images to complete and integrate concepts reported in the text. Thus, we added an example of NA detected by IVUS (now Figure 2 A in the new version of the review).

  1. Part 4.2. Optical Coherence Tomography (OCT) rows 199-250 The same, very important comment, the pictures are missing. They are very important in this kind of review.

Again, according to the Reviewer’s comment, we included a new figure showing NA using OCT (now Figure 2 B in the revised version of the manuscript).

  1. General comment about intravascular imaging parts. Provide a table considering the most important IVUS and OCT investigations about NA.

We added a table summarizing the most important IVUS and OCT investigations on NA (now Table 3 in the new version of the paper). Thanks for the comment. We also included the cited studies in the references list.

  1. 1.1. Lipid-lowering therapy rows 252-270. We don’t know whether guidelines recommendation for high-risk cardiovascular events for LDL level (LDL<1,4 mmol/L) can influence very late stent failure. And that is the basic question regarding the lipid-lowering therapy for the NA prevention and late and very late stent failure.

We agree with the Reviewer regarding the lack of specific guidelines recommendations to prevent NA with lipid-lowering therapy. We discussed this point in the revised version of the manuscript (page 9, lines 325-330).  We also included a new reference (now reference 70 in the re: Meng L, Liu X, Yu H, vised version of the manuscript): Wei G, Gu X, Chang X, Xie M, Qu W, Peng X, Sun Y, Tian J, Yu B. Incidence and Predictors of Neoatherosclerosis in Patients with Early In-Stent Restenosis Determined Using Optical Coherence Tomography. Int Heart J. 2020 Sep 29;61(5):872-878. doi: 10.1536/ihj.20-139. Epub 2020 Sep 12. PMID: 32921669.

  1. 1.2. Antiplatelet therapy. The type of stent used for PCI is only one of the factors which can influence the rate of very late stent failure. The complexity of PCI and the importance of stents' position are very important. Now I’m prone to favorite long term dual platelet or use of stronger P2Y12 blockers alone for complex PCI with clinical features of higher risk (diabetes, poor control of LDL, higher Lp(a), smokers, etc), with lower risk for major bleeding (patients younger than 75, and without a history of gastrointestinal or other diseases which could provoke bleeding).

We are aware of the complexity of clinical and procedural factors that influence the in-stent thrombotic risk in patients receiving coronary stenting, and we agree with the Reviewer that recommendations for the type and duration of antiplatelet therapy should take into account all these variables in order to prevent thrombosis, avoiding an excessive bleeding risk. As suggested by the Reviewer, we discuss this point in the revised version of the manuscript (page 10, lines 341-346 and 349-351).

  1. Is there any manuscript about the long-term use of P2Y12 blockers (longer than 3 years) in patients with stents? We are witnesses of stent thrombosis after 10 and more years now. The stents are locus resistensiae minoris for life.

According to the Reviewer’s comment, we added some evidence regarding the long-term use of P2Y12 inhibitors (> 3 years) after coronary stenting suggesting some benefit in the prevention of major adverse events (page 10, lines 351-357) (references 76-78 in the new version of the manuscript).

  1. 1.3. Targeting inflammation and oxidative pathways, rows 290-358. This is an un-appropriately long part with full of experimental and small clinical findings regarding prevention of in/stent restenosis, and not NA or very late thrombosis which is the main consequence of NA and the main topic of this review.

Chronic inflammation significantly contributes to NA development (Otsuka F, et al. Neoatherosclerosis: overview of histopathologic findings and implications for intravascular imaging assessment. Eur Heart J. 2015;36(32):2147-59; Nakazawa G, et al. The pathology of neoatherosclerosis in human coronary implants bare-metal and drug-eluting stents. J Am Coll Cardiol. 2011;57(11):1314-22; Otsuka F, et al. Pathology of second-generation everolimus-eluting stents versus first-generation sirolimus- and paclitaxel-eluting stents in humans. Circulation. 2014;129(2):211-23; Ochijewicz D, et al. Inflammation as a determinant of healing response after coronary stent implantation. Int J Cardiovasc Imaging. 2021;37(3):791-801). Thus, considering the pathophysiological mechanisms of NA, we reported in this session anti-inflammatory and anti-oxidative drugs that potentially might prevent this phenomenon since their specific mechanisms of action. We are aware that most of them have not been proven in the specific setting of NA, but late IRS represents one clinical manifestation of NA together with late and very late stent thrombosis. Thus, indirectly we could speculate that their benefit on late ISR might occur also through the delay and prevention of NA. However, according to the Reviewer, we revised this part of the manuscript removing unnecessary concepts (page 10, lines 370-387 in the new version of the paper).

  1. 2.1. Drug-eluting balloons and repeat drug-eluting stent implantation rows 360-379. The problems for all these trials are a relatively short follow-up, soft endpoints as outcomes, and the relatively small number of patients and we must be very cautious with the interpretation of clinical values of these studies. This critical approach to the results is lacking in the text.

We thank the Reviewer for this important point. We agree with him/her that studies investigating drug-eluting balloons and drug-eluting stents in the setting of NA included small populations with a relatively short follow-up. For this reason, we commented on this in the revised version of the manuscript (pages 11, lines 420-424 of the revised version of the manuscript).

  1. 2.2. Debulking strategies. As the authors stated, debulking techniques are not widely available and in spite of initially larger angiographic gain this is not reflecting the clinica l benefit and the place for this technique is far from clinical significance. Too much space for this.

We are aware that these invasive strategies have been investigated in small studies and have not been associated with better clinical outcomes notwithstanding an immediate angiographic post-procedural success. However, one of the main purposes of this review is to provide a comprehensive overview of all potential strategies in preventing and treating NA, thus we believe that all these strategies should be mentioned despite limited and conflicting data, possibly, as the Reviewer suggests, with a critical approach. However, according to the Reviewer’s comment, we revised this part of the manuscript removing unnecessary and redundant concepts (page 11, lines 426-438 of the revised version of the manuscript).

  1. 2.3. Bioresorbable vascular scaffolds. BVS should be used only as a part of clinical studies in patients with in-stent restenosis, and we must wait for the next generation of BVS to go ahead.

Thanks for the comment. We completely agree with the Reviewer. For this reason, we stated in the paragraph addressing BVS in this revised version of the manuscript that “Despite being currently an off-label strategy evaluated in selected small studies, BVS showed encouraging results regardless of ISR pattern or stent type, whereas long-term clinical outcomes are still a matter of debate” and “data supporting the routine adoption of BVS as an alternative tool for ISR treatment are currently lacking” (page 12, lines 457-460 and lines 466-467 of the revised version of the manuscript).

  1. Figure 2. Almost all these treatment options are lacking evidence-based data for use in this setting and some of them are totally in the experimental phase, it should be stated.

We thank the Reviewer for his/her comment. In this revised version of the manuscript, we clearly stated that all strategies proposed in Figure 2 might have a potential benefit on NA prevention and treatment since most of them are in the experimental phase and need confirmation in future studies (page 12, lines 468-470). Furthermore, we marginally modified Figure 2, including the concept of late stent thrombosis, since NA is the pathophysiological substrate of both late stent thrombosis and late stent restenosis. We believe that the new proposed Figure 2 is more complete than the previous one.

  1. General comment: do we have some investigation with long-term follow-up of stented patients and the incidence of NA in consecutive patients?

According to the Reviewer’s suggestion, we provided some evidence from observational registries regarding the incidence of NA at long-term FUP in multiple settings (either progressive ISR and VLST). Additionally, we briefly discussed the potential impact on MACE of different NA patterns, although large, randomized, clinical trials assessing the effect of neo-intimal classification on long-term clinical outcomes are currently lacking (page 7, lines 255-261 in the revised version of the manuscript, new references 55-57).

  1. In my opinion, this review article is lacking more pictures, tables, and information. Some parts are overwritten and some of them need more data. Conclusion: The article needs major revision to be considered for publication in life.

We believe in this revised version of the manuscript to have fully addressed the many helpful comments provided by the Reviewer and we hope now he/she may consider the paper for publication in Life.

Reviewer 2 Report

In the current manuscript Nusca et al. provide an overview of the pathophysiological processes related to instent restenosis, in particular the late stage in stent neo-atherosclerotic processes. This review paper gives an excellent overview of the processes and mechanisms underlying the complex phenomenon, as well as the clinical inplacations and therapeutic options. It is well written and logically structured and helps the reader to understand this interesting phenomenon.

However, I do have one small comment :

Figure 1 : the legends for this figure should be extended. In the current status the figure is difficult to comprehend. Especially the phenomena ’myointimal hyperplasia’ and ‘impaired re-endothelialisation’ are not discussed , or not referred to in the text.

Author Response

In the current manuscript Nusca et al. provide an overview of the pathophysiological processes related to in-stent restenosis, in particular the late stage in stent neo-atherosclerotic processes. This review paper gives an excellent overview of the processes and mechanisms underlying the complex phenomenon, as well as the clinical implications and therapeutic options. It is well written and logically structured and helps the reader to understand this interesting phenomenon. However, I do have one small comment.

We thank for the positive comments from the reviewer.

  1. Figure 1 : the legends for this figure should be extended. In the current status the figure is difficult to comprehend. Especially the phenomena ’myointimal hyperplasia’ and ‘impaired re-endothelialisation’ are not discussed , or not referred to in the text.

According to the Reviewer’s comment, we modified the legend of Figure 1, mostly using terms and definitions reported in the main text. Thanks for the comment.

Round 2

Reviewer 1 Report

The authors reply completely to my comments. I would like to thank them because of this great effort. I assume that the new version of MS is much better.